# Quantifying heterogeneities in arbovirus transmission: Description of the rationale and methodology for a prospective longitudinal study of dengue and Zika virus transmission in Iquitos, Peru (2014–2019)

Amy C. Morrison[1]*, Valerie A. Paz-Soldan[2], Gonzalo M. Vazquez-Prokopec[3], Louis Lambrechts[4], William H. Elson[5], Patricia Barrera[5,6], Helvio Astete[7,8], Veronica Briesemeister[5], Mariana Leguia[6], Sarah A. Jenkins[7], Kanya C. Long[9], Anna B. Kawiecki[1], Robert C. Reiner, Jr.[10], T. Alex Perkins[11], Alun L. Lloyd[12], Lance A. Waller[13], Robert D. Hontz[7], Steven T. Stoddard[14], Christopher M. Barker[1], Uriel Kitron[4], John P. Elder[14], Alan L. Rothman[15], Thomas W. Scott[5], on behalf of the Proyecto Dengue Group[¶]

1 Department of Pathology, Microbiology, and Immunology, School of Veterinary Medicine, University of California, Davis, Davis, California, United States of America, 2 Department of Tropical Medicine, Tulane University School of Public Health and Tropical Medicine, New Orleans, Lousiana, United States of America, 3 Department of Environmental Sciences, Emory University, Atlanta, Georgia, United States of America, 4 Institut Pasteur, Université Paris Cité, CNRS UMR2000, Insect-Virus Interactions Unit, Paris, France, 5 Department of Entomology and Nematology, University of California Davis, Davis, California, United States of America, 6 Genomics Laboratory, Pontificia Universidad Católica del Peru, Lima, Peru, 7 Virology and Emerging Infections Department, United States Naval Medical Research Unit No. 6, Lima, Peru, 8 Department of Entomology, United States Naval Medical Research Unit No. 6, Lima, Peru, 9 Department of Family Medicine and Public Health, University of California San Diego School of Medicine, La Jolla, California, United States of America, 10 University of Washington, Seattle, Washington, United States of America, 11 Department of Biological Sciences and Eck Institute for Global Health, University of Notre Dame, Notre Dame, Indiana, United States of America, 12 Biomathematics Graduate Program and Department of Mathematics, North Carolina State University, Raleigh, North Carolina, United States of America, 13 Department of Biostatistics and Bioinformatics, Rollins School of Public Health, Emory University, Atlanta, Georgia, United States of America, 14 School of Public Health, San Diego State University, San Diego, California, United States of America, 15 Institute for Immunology and Informatics and Department of Cell and Molecular Biology, University of Rhode Island, Providence, Rhode Island, United States of America

¶ Membership of the Proyecto Dengue Group is provided in the Acknowledgments.
* amy.aegypti@gmail.com, acmorrison@ucdavis.edu

## Abstract

Current knowledge of dengue virus (DENV) transmission provides only a partial understanding of a complex and dynamic system yielding a public health track record that has more failures than successes. An important part of the problem is that the foundation for contemporary interventions includes a series of longstanding, but untested, assumptions based on a relatively small portion of the human population; i.e., people who are convenient to study because they manifest clinically apparent disease. Approaching dengue from the perspective of people with overt illness has produced an extensive body of useful literature. It has not, however, fully embraced heterogeneities in virus transmission dynamics that are increasingly recognized as key information still missing in the struggle to control the most

**Data Availability Statement:** THIS MANUSCRIPT DOES NOT INCLUDE ANY UNDERLYING DATA SET AND, THEREFORE, DOES NOT INCLUDE ANY ANALYSIS OF DATA. IT IS LIMITED TO DESCRIPTION OF THE METHODOLOGY RATIONAL FOR OUR STUDY.

**Funding:** This study was funded by the U.S. National Institute of Allergy and Infectious Diseases (NIH/NIAID) award number P01 AI098670 (TWS). ACM is supported by award U01AI151814 from the National Institute of Allergy and Infectious Diseases of the National Institutes of Health. Further support was provided by Bill and Melinda Gates Foundation (BMGF) to the University of Notre Dame (Grant# OPP1081737) (NA), the Defense Threat Reduction Agency (DTRA) (RDH), Military Infectious Disease Research Program (MIDRP, S0520_15_LI and S0572_17_LI), and the Armed Forces Health Surveillance Branch Global Emerging Infections Systems research program (GEIS) ProMIS ID: 20160390169, P0090_17_N6_1.1.1, P0106_18_N6_01.01, and P0143_19_N6.

**Competing interests:** The authors have declared that no competing interests exist.

important insect-transmitted viral infection of humans. Only in the last 20 years have there been significant efforts to carry out comprehensive longitudinal dengue studies. This manuscript provides the rationale and comprehensive, integrated description of the methodology for a five-year longitudinal cohort study based in the tropical city of Iquitos, in the heart of the Peruvian Amazon. Primary data collection for this study was completed in 2019. Although some manuscripts have been published to date, our principal objective here is to support subsequent publications by describing in detail the structure, methodology, and significance of a specific research program. Our project was designed to study people across the entire continuum of disease, with the ultimate goal of quantifying heterogeneities in human variables that affect DENV transmission dynamics and prevention. Because our study design is applicable to other *Aedes* transmitted viruses, we used it to gain insights into Zika virus (ZIKV) transmission when during the project period ZIKV was introduced and circulated in Iquitos. Our prospective contact cluster investigation design was initiated by detecttion of a person with a symptomatic DENV infection and then followed that person's immediate contacts. This allowed us to monitor individuals at high risk of DENV infection, including people with clinically inapparent and mild infections that are otherwise difficult to detect. We aimed to fill knowledge gaps by defining the contribution to DENV transmission dynamics of (1) the understudied majority of DENV-infected people with inapparent and mild infections and (2) epidemiological, entomological, and socio-behavioral sources of heterogeneity. By accounting for factors underlying variation in each person's contribution to transmission we sought to better determine the type and extent of effort needed to better prevent virus transmission and disease.

## Introduction

Despite more than a century of intensive efforts, development of effective strategies to prevent *Aedes*-transmitted viral disease continues to lag behind the burden of disease, in part because of a paucity of empirical data examining pathogen transmission dynamics [1]. Interventions are largely founded on untested assumptions derived from observations of a relatively small portion of the human population, i.e., people who are convenient to study because they are ill and seek medical treatment. Evaluations of dengue from the perspective of people with overt illness have produced an extensive body of valuable literature. Those studies do not, however, capture heterogeneities in virus transmission dynamics that are increasingly recognized as key to controlling the most important insect-transmitted viral infection in humans. Only in the last 20 years have there been significant efforts to carry out comprehensive longitudinal dengue studies [2, 3] that allow research questions to be broadened to include a larger portion of the dengue virus (DENV)-infected human population.

Herein, we present the rationale, goals, and study design of a longitudinal research program entitled "Quantifying Heterogeneities in Dengue Virus Transmission" conducted in Iquitos, Peru from July 2014 to April 2019. This program was an extension of research conducted between 2006–2013 that examined the contribution of human movement to DENV transmission dynamics and included new expertise in behavioral science, spatial analysis, and modeling [4–9]. This publication describes our large and complex research program that followed our human movement studies. It is intended to be a single unified source that can be referenced in subsequent publications. Our aim is to provide a holistic overview of the integrated structure

and methodology that we used to achieve our research objectives. Data collection for the project ended in 2019. Due to delays associated with the COVID-19 pandemic some data analysis and laboratory testing are ongoing. Although we previously published project associated reports (i.e., spectrum of dengue illness experienced in Iquitos [10] and impact of dengue on human mobility [11, 12] and quality of life measures [13]) in this publication we do not focus on project related data. Instead, we limit our presentation to the overall rationale and methodology of the study and how the project was designed to directly address key knowledge gaps in the understanding of DENV transmission.

## Study rationale

Our study builds on findings from a series of longitidunal cohort studies conducted between 1999 and 2014, which showed that human movement accounts for significant variation in the spatial spread of DENV, that few individuals or locations may be giving rise to a disproportionate number of new infections, and the proportion of inapparent infections was high [2, 4, 14]. All of these factors are principal contributors to DENV transmission and could constitute strategic targets for intervention. Thus, a critical next step was to define the contribution of an understudied portion of total DENV infections, the estimated 300 million people [15] with inapparent and mild ambulatory infections. to DDENV transmission dynamics. Specifically, we tested the hypothesis that people infected with DENV vary in their contributions to transmission as a function of their viremia [16, 17], activity patterns, and contact [4] with mosquitoes. We predicted that people with inapparent and mild DENV infections contribute more to DENV transmission than previously recognized because these individuals are infectious to mosquito vectors and are less likely to limit their mobility due to their symptoms.

To test this prediction, it was essential that we identify people with active DENV infections across the spectrum of disease, especially those with inapparent and mild infections, who are not normally detected by disease surveillance systems. We built on an established multi-layered approach to real-time detection of human DENV infections that leveraged (1) a joint U.S. Naval Medical Research Unit No. 6 (NAMRU-6) and Peruvian Ministry of Health clinic-based surveillance program, (2) active community-based surveillance, and (3) contact cluster investigations for efficient detection of additional DENV infections, including people with inapparent and mild disease. By quantifying heterogeneities in people's individual contributions to DENV transmission dynamics, we aimed to fill important knowledge gaps leading to improved strategies to prevent DENV transmission and disease. Ultimately, our study design and results could extend to research and prevention efforts for other infectious diseases.

## Significance

DENV causes more human morbidity and mortality worldwide than any other arthropod-borne virus and has expanded its distribution further than any other vector-borne disease [15, 18–21]. There are as many as 4 billion people living in areas currently at risk of infection, with 400 million estimated new infections and 50–100 million new symptomatic infections occurring each year [15, 21, 22]. From 1990 to 2013, dengue cases increased 4-fold and the virus rapidly increased its geographic range causing disease in new exposed susceptible populations [15, 21–24]. In most of the world, dengue control is failing, despite the fact that some forms of vector control are theoretically straightforward. Unsuccessful programs are often attributed to a lack of resources, lack of political will, ineffective implementation [1, 25], and/or the absence of a rigorous evidence base for available methods that was generated through robust study designs with epidemiological endpoints [26, 27]. Equally important is our incomplete understanding of (1) the relationships between human behavior, mosquito ecology, and virus

transmission dynamics; (2) the most appropriate methods to assess and respond to risk; and (3) the most effective use of surveillance information to inform programmatic control decisions [28]. Fundamental concepts of dengue disease prevention remain incompletely defined and underutilized. New theories, concepts, and tools are needed to improve dengue prevention. To this end, our program was designed to assess the contribution of the understudied majority of DENV-infected people (i.e., those with clinically inapparent/mild infections) to the epidemiological, entomological, and socio-behavioral heterogeneities in DENV transmission dynamics. Longitudinal cohort studies only detect clinically inapparent/mild infections in small numbers [29, 30] or retrospectively after their infectious, viremic period ended [2, 30]. Our contact cluster protocol can identify people with mild disease manifestations while they are viremic. We, therefore, shifted our research focus to people who do not manifest debilitating illness, but potentially are critical contributors to the temporal and spatial dynamics that amplify DENV transmission and whose infections confound prediction of explosive dengue outbreaks.

A key component of the heterogeneity in dengue epidemiology is the well-established phenomenon that typically half or more of people with DENV infections are not detected by standard public health surveillance [2, 20, 30–32]. Variation in the ratio of symptomatic to inapparent/mild infections is often estimated retrospectively in longitudinal cohort studies [2, 3, 31], but the contribution of people with inapparent/mild infections to virus transmission remains largely unexplored. Overlooking inapparent/mild infections can result in (as has been seen for cholera) [33] inapparent infections forcing a more rapid rise and more rapid subsequent fall of epidemics, shifting the perceived peak of the epidemic curve to an earlier time period than was previously recognized, and accelerating the process of pathogen epidemic transmission and geographic spread. This has become especially apparent globally in 2020. SARS-CoV-2 transmission dynamics similarly appears to be fueled by innapparent and mild infections [34], underlining the importance of this understudied segment of the virus transmission process.

Whether inapparent/mild DENV infections produce a similar pattern to the one observed for cholera or SARS-CoV-2 remains to be determined. What is clear, however, is that the rapid spatial spread of epidemic dengue cannot be explained by clinically apparent infections alone [35]. For example, identification of seroconversions of a longitudinal cohort compared to virus isolates from a clinic-based surveillance study in Iquitos detected a 5–6 month period of "silent" transmission prior to the 2002 DENV-3 invasion into the city [2]. During 2008, DENV-4 circulated from February-July virtually unnoticed before causing a large outbreak later that same year [36, 37]. During the largest dengue epidemic in Iquitos history during 2010–2011 (DENV-2 Asian-American strain), severe disease was detected when attack rates were highest. These observations imply a functional relationship between force of infection and observed cases of severe disease with a prominent role played by people with inapparent/mild infections in virus transmission dynamics.

We expect that results from our project will have public health and basic science implications. If people across the continuum of disease are more accurately accounted for (1) dynamics in virus transmission will be more accurately interpreted, (2) surveillance and triggers for implementing control will be more effectively timed for maximum impact, (3) targets for vaccine and vector control evaluation and delivery will be more successfully estimated and implemented, (4) immunological precursors of dengue severity can be studied across a broader range of disease outcomes, and (5) data across all disease manifestations can be included in comprehensive analyses (mathematical and simulation models) of dengue transmission and control. Almost all dengue endemic countries base their surveillance on clinically apparent dengue cases and consequently miss the majority of infections. We hope that our results will

encourage development of capacity to respond more quickly and, thus, more effectively to elevated risk and will drive innovations for quick, accurate, and inexpensive detection of a greater portion of infections. We expect our research will be of interest to people evaluating dengue vaccines and vector control, particularly if goals include dengue elimination (i.e., preventing the few remaining infections from spawning a resurgent wave of disease) and the prevention of re-emergence.

## Program plan

Our overall goals were to (1) quantitatively define the relationship between severity of disease and contribution to DENV transmission dynamics; (2) assess the relative and joint contributions of epidemiological, entomological, and socio-behavioral sources of heterogeneity to the dynamics of DENV transmission; and (3) predict risk to identify public health measures to minimize DENV transmission and disease. Three distinct, but interdependent, projects shared a Surveillance Core and were supported by Administrative and Data Cores (Fig 1). Below, we review the structure and methods for projects and cores beginning with the Surveillance Core, followed by the three projects, and concluding with the Data and Administrative Cores. Although our study focus was dengue, from June 2016 to April 2017 Zika virus circulated in Iquitos, providing us the opportunity to ask the same questions for both viruses.

### Surveillance core

The Surveillance Core was designed to identify individuals with active DENV infections for detailed study in Projects 1 and 2 (Fig 2, see S1 Appendix for detailed methods). Activities encompassed field surveillance operations to identify acute disease cases to estimate disease incidence, routine serological (annual longitudinal blood samples from a pediatric subset of surveillance participants to estimate DENV/ZIKV seroconversion rates), routine

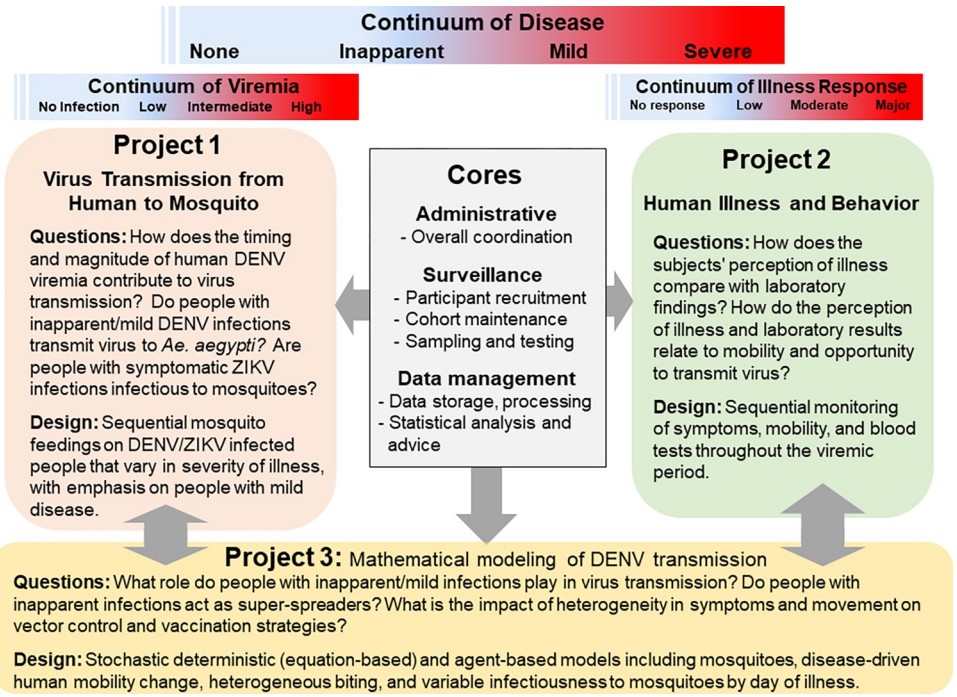

**Fig 1. Project diagram showing the integration of the three projects and cores.**

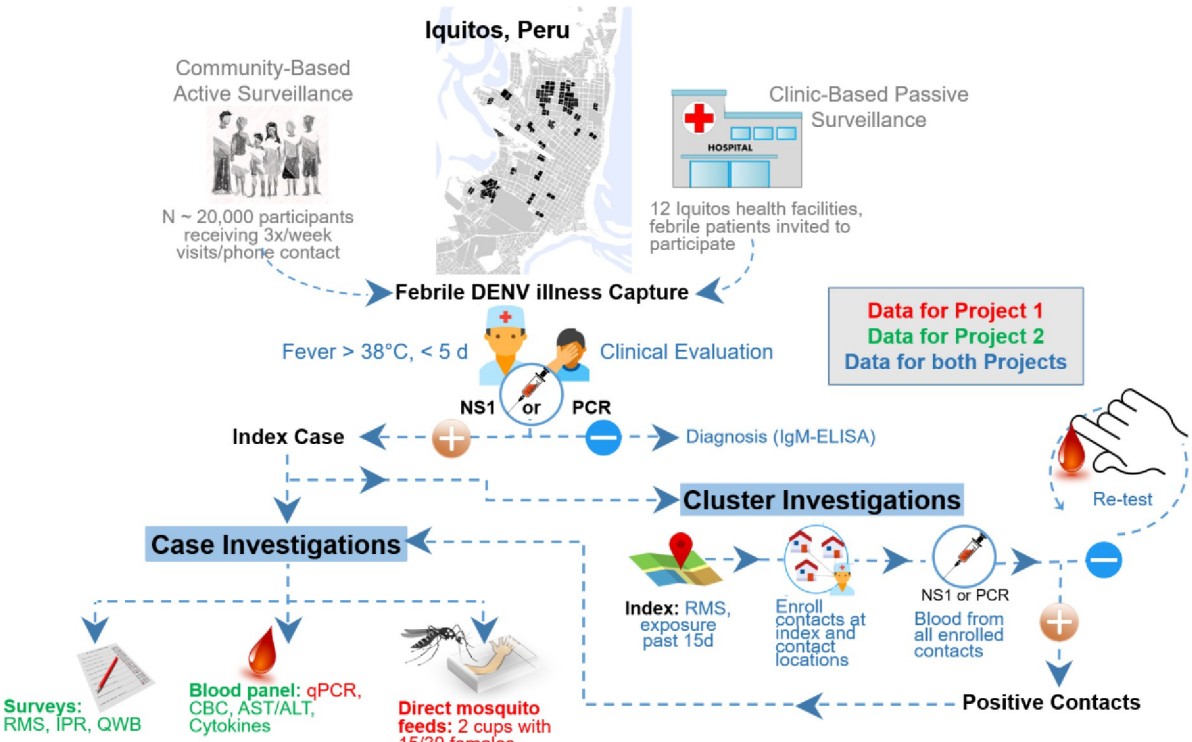

**Fig 2. Summary of procedures followed to collect individual data on human infectiousness to mosquitoes (Project 1) and their response to dengue and Zika illness (Project 2).** Participant capture occurred through multiple sources, including community-based surveillance, hospital-based surveillance, and contact-cluster investigations. Once viremic individuals were identified, sequential data on their behavior (retrospective mobility survey, RMS; infection-perception and response, IPR; and quality of well-being, QWB) and markers of infection in blood [viremia titer by qPCR, Complete Blood Count CBC, Apartate transaminase (AST)/Alanine aminotransferast (ALT), and cytokines], and direct laboratory-reared mosquito feeding (using 30 female mosquitoes for children and 60 mosquitoes for adults) were conducted throughout the illness period of each participant.

entomological monitoring, and laboratory procedures for diagnostic and project-specific purposes. We took a proactive and multi-layered approach to surveillance. Cluster investigations were initiated by febrile individuals with detectable DENV or ZIKV viremias who reported the locations they visited in the previous 15 days. Susceptible contacts at reported residential locations were monitored for signs of infection over an ~30 day period. Infections were identified by NS1 antigen detection and RTq-PCR within 24 hours of blood collection. As soon as a viremic individual was detected, that person was recruited to participate in mosquito feeding experiments (Project 1) and monitoring of symptoms and behavior (Project 2). This nested approach to case detection facilitated direct comparison between alternative surveillance schemes and validation of models developed in Project 3.

### Project 1: Dengue and Zika virus transmission from viremic people to mosquitoes

Project 1 was designed to quantitatively define the relationship between an individual's infection across the spectrum of disease (inapparent to severe) and their contribution to DENV transmission. *Aedes aegypti* from a confirmed uninfected genetically diverse laboratory strain (GDLS) were fed directly on NS1/PCR-positive DENV-infected people (30 [for children] - 60 [for adults] mosquitoes per participant) (Section 3.3 in S1 Appendix). Rates of mosquito infection and *in vitro* transmission (i.e., assay of expectorated saliva) were determined (Fig 3,

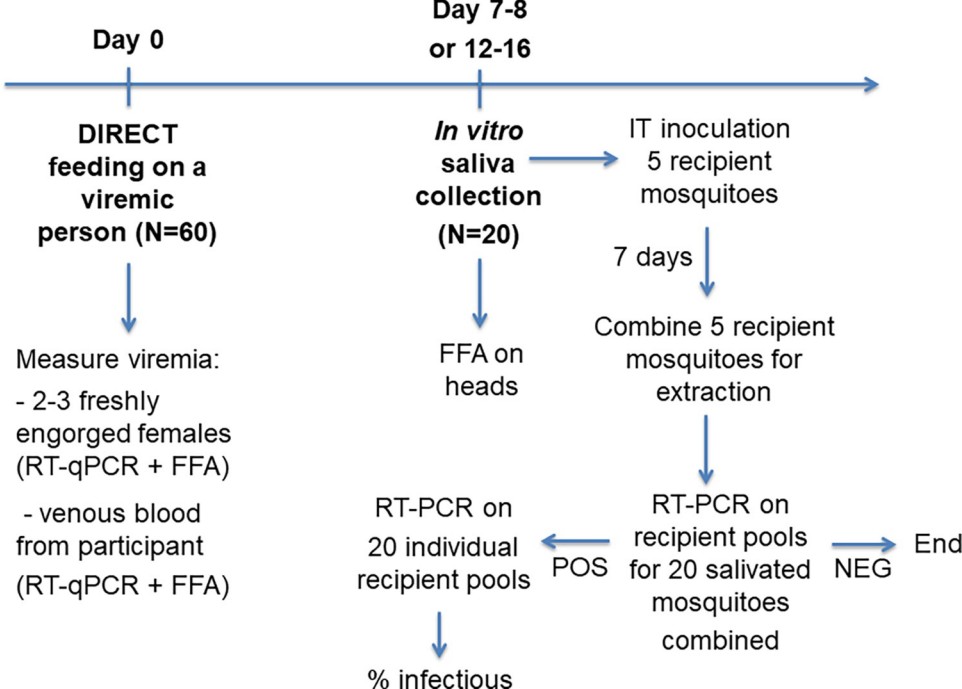

**Fig 3. Workflow for mosquito feeding procedures carried out on DENV/ZIKV infected study participants.**
FFA = Fluorescent Focus Assay, RT-PCR = Reverse Transcriptase Polymerase Chain Reaction.

Section 5 in S1 Appendix). This allowed us to define the capacity of people with different disease manifestations and viremia levels to transmit virus to the mosquitoes they encountered during their daily activities (Project 2) and characterize the relative contributions of people with different dengue illnesses to seasonal and epidemic dengue dynamics (Project 3).

## Project 2: Effects of dengue or Zika virus infection on human behavior with virus transmission potential

Project 2 assessed the relative and joint contributions of epidemiological, social, and human behavioral sources of heterogeneity to the dynamics of DENV transmission (Fig 2). Data were collected through interviews (Table 1, Section 6 in S1 Appendix) and laboratory analyses of blood samples from individuals with acute viremic DENV infections (Sections 4.1–4.4 in S1 Appendix). These were the same individuals participating in the mosquito feeding experiments (Project 1). By applying social and health science techniques we (1) described and quantified the association between subjective and objective measures of dengue disease severity, (2) examined and quantified human behavior changes associated with dengue disease of various severities [10, 11, 13], and (3) estimated individual human-mosquito contact rates to explore risk for potential forward DENV transmission (Project 3).

## Project 3: Drivers of heterogeneities in dengue virus epidemiology, transmission dynamics, and control

Project 3 aimed to predict risk and identify the most effective public health measures to minimize virus transmission and disease. To quantify and assess the impact on dengue virus

**Table 1. Interview workflow for 8 survey types using surveys developed for CommCare [41] and deployed on mobile tablets running Android OS [51], each questionnaire was administered verbally by trained research staff, who recorded their responses using the application.** Shaded cells denote when surveys were applied.

| Timeline | 8 survey types | | | | | | | |
|---|---|---|---|---|---|---|---|---|
| | Consent (written) | Full RMS[1] | 24hr RMS[2] | IPR[3] | HRQoL[4] | Costs[5] | Blood Sample[6] | GPS[7] |
| Enrollment day- Index case captured and tested by PCR or NS1 | �acb | ▓ | | ▓ | ▓ | ▓ | ▓ | ▓ |
| Day 1 | | | ▓ | ▓ | | | ▓ | ▓ |
| Daily-subsequent days | | | ▓ | ▓ | | | ▓ | ▓ |
| Day 3 | | | | ▓ | | | | |
| Last day (after PCR) | | | ▓ | ▓ | | | ▓ | ▓ |
| One month follow-up | | ▓ | | ▓ | | ▓ | ▓ | |
| Enrollment day-Contact | ▓ | | | ▓ | | | ▓ | |
| Follow up with contact NOT PCR+ (every 3–7 d) | | | | ▓ | | | ▓ | |
| Contact after first PCR + | ▓ | ▓ | | ▓ | | | ▓ | |

[1]Two-week Retrospective Movement Survey (places visited by participant in last 15 days)

[2]Daily Retrospective Movement Survey (places visited since last seen by study team)

[3]Illness Perception and Responses survey

[4]Health-Related Quality of Life survey

[5]Costs = short economic survey (family expenditures)

[6]Blood sample, refers to request, even through a sample was not always provided)

[7]GPS provided to the participant.

transmission dynamics and the effectiveness of different surveillance and control scenarios, we drew on empirical field data from Projects 1 and 2 and our previous research in Iquitos and generated mechanistic mathematical models that account for (1) host and vector movement, (2) a realistic spatial structure and mosquito population dynamics, and (3) heterogeneities in human infectiousness, attractiveness to mosquito bites, and mobility behavior as a function of disease severity.

## Data core

The Data Core upgraded data support operations that were developed over the previous 20 years of dengue-related research in Iquitos. Key improvements were (1) implementation of a more advanced relational database in PostgreSQL [38] that integrated historical and current data, (2) implementation of open-source GIS tools built on PostGIS [39] and managed with QGIS [40], (3) development of a CommCare [41] mobile data collection system for collecting data from household surveys using tablets and mobile phones, (4) development of a secure, web-based user interface using the Django framework [42] for daily data management in collaboration with the University of Notre Dame's Center for Research Computing (as a component of a clinical trial evaluating a spatial repellent in Iquitos [43], and (5) establishment of regular data redundancy and backup processes across multiple servers in Iquitos and UC Davis to ensure data preservation and integrity. The Data Core worked closely with the Surveillance Core to manage the project's data systems, train project personnel as needed, and distribute and exchange data among the research team and key partners, such as NAMRU-6.

## Administration core

The Administrative Core provided overall fiscal and research administrative support, including budget, fiscal, and progress reporting and IRB/human subjects compliance. The Administrative Core was based at the lead institution (UC Davis) and coordinated with local direction at the Iquitos field site.

## Ethical considerations and consent

The study protocol was approved by the U.S. Naval Medical Research Unit No. 6 Institutional Review Board (Protocol Number NAMRU6.2014.0028), which included Peruvian representation and complied with US Federal and Peruvian regulations governing the protection of human subjects. IRB authorization agreements were established between U.S. Naval Medical Research Unit No. 6 and all participating institutions. The protocol was reviewed and approved by the Loreto Regional Health Department (LRHD), which oversaw health research in Iquitos. A clinic-based surveillance protocol (NMRCD2010.0010) and ongoing community-based vector control intervention protocol (NAMRU6.2014.0021) identified DENV/ZIKV positive cases that could be invited to participate in virus positive case protocols and initiate cluster investigations. The study had an information sheet that explained the overall goals and the different components of the study. Non-invasive components of the study, such as a household census, surveillance visits from study personnel, mosquito surveys, focus groups, and some pilot studies required only verbal consent. The study had written consent documents for the following separate activities: (1) providing annual blood samples for longitudinal monitoring of DENV neutralizing antibodies in child participants, (2) providing acute and convalescent blood sample and movement survey in the case of illness (samples tested for DENV/ZIKV), (3) participation in a cluster contact investigation (up to 5 blood samples in 30 days), and (4) as a DENV/ZIKV positive case that provided permission for all the procedures in Fig 2 and Table 1. Participants that were minors provided assent (written for 8-17-years old, verbal for <8 years old). Their parents or guardians provided written consent. Consent without written documentation (verbal) was approved for non-invasive procedures and activities including census, entomological surveys, focus group discussions, and some noninvasive pilot studies.

## Study area

All study activities were carried out in Iquitos, a geographically isolated urban center approaching 400,000 inhabitants [44], located in the Amazon Basin of northeastern Peru. The city is located on a peninsula, surrounded on three sides by major rivers (Itaya, Nanay, and Amazon), and functions as a natural socio-epidemiological system because access to it is limited to air and river travel. The U.S. Navy initiated epidemiological studies on DENV in Iquitos in the early 1990s, then partnered with the University of California at Davis to study sequential invasions of distinct DENV serotypes. In general, each year DENV activity is low from May to July, increases in August and September, and then usually peaks between November and January. If no vector control is implemented, elevated levels of transmission can continue through April [45, 46]. In May 2016, the first human ZIKV infections were reported in Iquitos. ZIKV transmission continued through April 2017 with almost no detection of DENV during that time period [47, 48].

Iquitos is composed of parts of four larger districts, Punchana to the north, San Juan Bautista to the south, Belen to southeast, and Iquitos that extends to the west, but includes the core of the city. The city has experienced rapid urbanization and expansion in the past three decades [49], from the neighborhoods around the city center/commercial zones in the districts of Iquitos and Punchana to areas on the river (Belen and parts of Punchana) and to the south (San Juan Bautista). Each district includes less developed areas on the outer edges of the city to smaller rural communities along the Iquitos-Nauta highway and surrounding rivers with well developed transportation networks into the city. The main industries are small commercial enterprises, extractive business (logging, mining), and agricultural.

Our study followed and identified DENV and ZIKV cases from two ongoing cohort studies, both initiated in June 2015, and carried out in the more developed and central parts of the

Iquitos and Punchana districts (Fig 2, S1 Fig in S1 Appendix full size map shown in Fig 2). Between June 2015-April 2019, ~20,000 participants were under disease surveillance. DENV/ ZIKV acute cases were recruited from these ongoing cohort studies and from an ongoing clinic-based passive surveillance study [36]. The cohort neighborhoods were relatively homogenous, with a patchwork of households ranging from wood structures with dirt floors to brick and concrete and ceramic floors. The Peruvian Statistics and Informatics Institute states that about 30% of the urban populations in Amazon regions of Peru have at least one unmet basic need, 18.2% live in poverty, and 3% live in extreme poverty [49]. Evidence of extreme wealth is not observed in Iquitos, and luxury items such as air conditioners are rarely observed. Other indicators for the Maynas and Punchana districts indicate that 93% of structures are individual houses (row houses with shared walls), 82–83% have corrugated metal roofs, 90–97% have electricity, and literacy rates are 88–92% [44].

## Study design

To achieve our program goals (Fig 2), our field team consisted of nurse technicians (n = 12– 20), entomological technicians (n = 15), a "movement team" (4 team members who managed cluster investigations and conducted project 2 interviews), a small data management team, and a group of physician and nurse supervisors. Nurse supervisors enrolled households in the study areas and conducted disease surveillance visits to all houses in the cohort 3–5 times per week to identify acute DENV/ZIKV infections. We modified our protocol after the invasion of ZIKV into Iquitos during 2016 to include DENV and ZIKV-specific case definitions and differential laboratory diagnosis. From a subset of pediatric participants (i.e., an embedded longitudinal component), we drew blood annually to detect DENV antibody seroconversions. People with acute disease detected in community-based and passive clinic-based disease surveillance, and through contact-cluster investigations, were invited to participate in mosquito feeding experiments and illness experience and behavioral (movement) studies, and asked to initiate contact-cluster investigations. These activities were the methodological backbone of the program and were driven by the Surveillance and Data Cores.

**Procedures performed on DENV- and ZIKV-infected participants.** Suspected and/or confirmed DENV/ZIKV cases from community- and clinic-based febrile surveillance or cluster contact investigations were invited to participate in (1) mosquito feeding procedures (Fig 3) and (2) collection of data related to their experience with illness and its intensity, health-related quality of life, and fine-scale history of movement. Participation consisted of allowing laboratory reared mosquitoes to feed directly on them (see Section 5.1 in S1 Appendix); collecting blood samples (EDTA and serum separator tube by venipuncture, see Section 2.1 in S1 Appendix) for complete blood count, cytokine, and liver enzyme testing; and responding to a series of seven separate surveys (Table 1). The procedures for conducting direct mosquito feeds on viremic participants were developed during a pilot study completed prior to the initiation of the program project [50]. Study procedures were repeated each day the participant's blood tested positive for DENV or ZIKV during the previous days' samples and the participant agreed to continue. Participants could decline individual procedures on to participate during individual days and remain in the study.

## Data integration and analysis plan

To support the scientific and operational aspects of the research program, we developed innovative approaches to the collection of information on an individual's infectiousness to mosquitoes and response to illness throughout the infectious period of each participant. All Projects and the Surveillance Core generated data and information that was managed by the Data Core

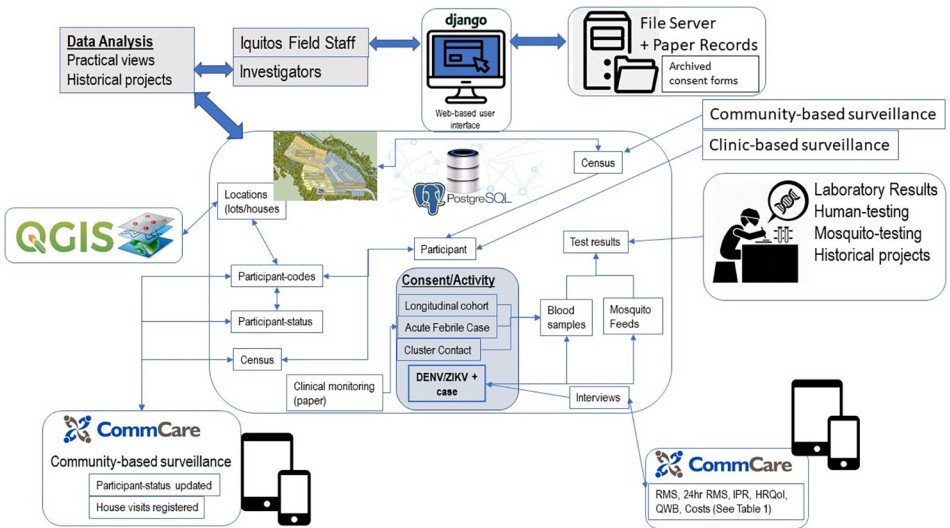

**Fig 4. Data structure and integration for the project.** All data was managed in PostgreSQL relational database with a direct link to a geographic information system in Quantum GIS. Additional data was captured through mobile android applications using CommCare software and later processed and imported into PostgreSQL.

in our secure database infrastructure (Fig 4). The Data Core provided real-time access to field personnel and investigators through a user-friend web-based Django interface. Data was entered by field staff either manually from paper forms or from downloaded data collected through android applications. Alternatively, the seven interview data collection instruments (Table 1) associated with dengue or Zika infected individuals were imported directly into the relational database. Each individual participant was registered in the system through a "census" form for an individual household. All individual data was geo-coded to the household level in a Geographic Information System (GIS). Individuals could only be added to locations found in the GIS. New locations were added constantly, either because they were newly added locations or households that split into two, after field validation with a GPS.

All individuals in a household received a "participant_code" that included the "location_code". For example, five people enrolled at the location_code MYC200 would be assigned the following participant_codes: MYC200P01, . . ., MYC200P05. On each cityblock the participant_codes were in alphanumeric order. If people, moved or houses were changed this information was managed in the participant_status table. Following the previous sample a year into the study house MYC200 was split in two houses (MYC200A and MYC200B), with the first two individuals staying in A and the remaining individuals in B. New participant_codes would be MYC200AP01 and MYC200AP02 and for house B (MYC200BP01, MYC200BP02, MYCB200P03 corresponding to MYC200P03-P05).

In the participant_status table each code had a start/stop date, except for the most recent status of the code. A status column kept track of active and inactive period. This information was used to calculate the person-time a person was under surveillance under any time interval specified.

The most innovative aspect of this data structure was the ability to following individuals from one location to another, or more importantly manage individuals who spent time in different locations during surveillance periods. That is, individuals could have more than one active participant_code at one time. The database system was duplicated on three servers, two in Iquitos and one in Davis. Ideally, the information was synchronized in realtime, but because of the bandwidth limitation in Iquitos, all data entry of human data occurred on one server

and entomological data on the other. Sychronization was carried out at least weekly intervals, and daily tape backs were made from the Davis server.

Although the primary unit of analysis was the participant, information was usually grouped through a "consent" table linking to individuals who participated in the DENV/ZIKV + case protocol (Fig 4, Table 1). All blood samples and mosquito feeds and their respective laboratory results were linked to the participant through the consent_id. For all of the listed activities each participant had multiple observations (i.e., multiple days throughout the infectious period) per participant. Given the time-dependent nature of our data, most statistical analyses included parameters that accounted for the individual variability in response.

Generalized Linear Mixed models (GLMM) will be used widely used to account for individual variability (random intercept, set on participant ID) and temporal variability (random slope, setting time as a time-varying factor when applicable) [52]. For Project 1, the time varying mosquito infectiousness data (dependent variable, binomial) is being analyzed using multiple models to identify its association with independent variables, such as, day of illness, viremia titer, and disease severity.

Using similarly structured time-varying data, Project 2 will quantify (1) the functional relationship between self-reported and clinical outcomes of disease, (2) identified the temporal variations in such relationships, and (3) assessed the composite contribution of factors to the overall perception of dengue severity.

## Programmatic synergies

Our project was designed to have a high-level of synergy between projects and cores in the flow of research aims, activities, data, and results (Figs 1 and 4). The Surveillance Core identified viremic people with varying disease manifestations for studies in Project 1 and 2 and provided data on the epidemiological context that helped define parameters for analyses carried out in Project 3; i.e., measures of herd immunity, community-wide dengue incidence rates, and fluctuations in Ae. aegypti population densities. Project 1 determined the transmission potential of viremic individuals, measured by relative infectiousness to mosquitoes that can be compared directly to objective and subjective measures of disease severity and behavior of those same individuals in Project 2. Project 2 estimated the risk of viremic people in Project 1 being bitten by mosquitoes. Projects 1, 2, and 3 required assimilation of data into a manageable format by the Data Core. The full impact of our project required the development of transmission and intervention models in Project 3 that are based on results from Projects 1 and 2, on concurrent epidemiologic trends provided by the Surveillance Core, and on historical patterns of DENV transmission provided by the Data Core. Information flowing from the Surveillance Core and all three projects was efficiently accumulated, processed, managed, integrated, and disseminated from the centralized Data Core. The Administrative Core provided administrative and financial oversight, logistical support, and outreach coordination for the other 2 cores and the 3 projects. We used a two-tiered administrative structure with one unit in Davis where the project PI, Program Manager, and Data Core were based and another unit in Iquitos where Morrison, the Surveillance Core Leader, lived and directed similar large integrative projects over the prior 14 plus years.

It is important to note that the composition of our research team was a blend of senior and junior scientists with integral participation by Peruvian scientists. We built the program on a history of outstanding collaboration among the research team, NAMRU-6 in Lima, and the Loreto Regional Department of Health (Ministry of Health). Our project and the research team structure provided a unique opportunity for junior researchers to continue to grow into key players in the fields of tropical disease ecology and epidemiology.

## Supporting information

**S1 Appendix. Supplementary appendix.** This appendix has been provided by the authors to give readers additional information about their work.
(DOCX)

## Acknowledgments

We thank the residents of Iquitos for their support and participation in this study. We greatly appreciate the support of the Loreto Regional Health Department, including Drs. Hugo Rodriguez-Ferruci, Christian Carey, Carlos Alvarez, Hernan Silva and the Lic. Wilma Casanova Rojas, who all facilitated our work in Iquitos. A special thanks to Gloria Talledo for her ongoing support with the preparation of IRB protocols and reports for this project. We appreciate the commentary and advice provided by the NAMRU-6 Institutional Review Board and Research Administration Program for the duration of this study.

A special thank you goes to our data management personnel (Gabriela Vasquez De La Torre, Magaly Ochoa Isuiza), and the nurse technicians involved in case capture (Llerme Armas Pisco, Cesar Augusto Banda Chavez, Linder Maria Canayo Zavaleta, Clara Chávez López, Karina Chuquipiondo Vasquez, Laury Dacia Cuespan Camus, Leny Curico Manihuari, Nadia Rocio Del Rio Chavez, Salome Elespuru, Junnelhy Mireya Flores López, Juan Flores Michi, Luz Angelica Galvez, Huayllahua, Rina Gonzales Jaba, Deisy Irene Huiñapi Cambunugue, Maria Edith Juárez Baldera, Xiomara Mafaldo García, Nora Marín Romero, Nadia Tereshkova Montes Criollo, Johnni Mozombite Flores, Sandra Ivonne Moñoz Perez, Lucy Navarro Sánchez, Geraldine Ocmin Galán, Zenith María Pezo Villacorta, Fiona Stefani Pinedo Zevallos, Iris Reátegui Carrión, Zoila Martha Reategui Chota, Sadith Jovita Ricopa Manuyama, Liliana Rios López, Rubiela Nerza Rubio Briceño, Ysabel Ruiz Berger, Miranda Angela Rocio Soplin, Rosana Magaly Sotero Jiménez, Rosa Tamani Babilonia, Zenith Tamani Guerrero, Moises Tanchiva, Sarita Del Pilar Tuesta Dávila, Flora Vargas Ceras, Rita Gabriela Vasquez Orbe).

The Proyecto Dengue Group is comprised of the following individuals. Jhonny Cordova (designated group lead, email: jhonnycordova777@gmail.com), Alfonso Vizcarra, and Arnold Noreiga affiliated with the Department of Entomology and Nematology, at the University of California, Davis. Jennifer E. Rios, W. Lorena Quiroz, S., Juan Sulca, Julia Sonia Ampuero, Isabel Bazan, Crystyan Siles, Stalin Vilcarromero, Leslye Angulo, Guadalupe Flores, Carolina Guevara, Maria Silva, Christopher Mores, Eric S. Halsey, Regina Fernandez, and Wesley R. Campbell are members of the Virology and Emerging Infections Department and Gissella Vasquez, from the Entomology Department of the U.S. Naval Medical Research Unit No. 6. Kathrine L. Schaber was in the Program of Population Biology, Ecology, and Evolution, Emory University.

**Disclaimer:** The views expressed in this article are those of the authors and do not necessarily reflect the official policy or position of the Department of the Navy, Department of Defense, nor the U.S. Government.

## Author Contributions

**Conceptualization:** Amy C. Morrison, Valerie A. Paz-Soldan, Gonzalo M. Vazquez-Prokopec, Louis Lambrechts, Kanya C. Long, Robert C. Reiner, Jr., T. Alex Perkins, Alun L. Lloyd, Lance A. Waller, Steven T. Stoddard, Uriel Kitron, John P. Elder, Alan L. Rothman, Thomas W. Scott.

**Data curation:** Amy C. Morrison, Gonzalo M. Vazquez-Prokopec, William H. Elson, Patricia Barrera, Veronica Briesemeister, Anna B. Kawiecki, Robert C. Reiner, Jr., Alan L. Rothman.

**Formal analysis:** Amy C. Morrison, Valerie A. Paz-Soldan, Gonzalo M. Vazquez-Prokopec, Louis Lambrechts, William H. Elson, Mariana Leguia, Robert C. Reiner, Jr., T. Alex Perkins, Alun L. Lloyd, Lance A. Waller.

**Funding acquisition:** Amy C. Morrison, Valerie A. Paz-Soldan, Gonzalo M. Vazquez-Prokopec, Louis Lambrechts, Robert C. Reiner, Jr., Alun L. Lloyd, Robert D. Hontz, John P. Elder, Alan L. Rothman, Thomas W. Scott.

**Investigation:** Amy C. Morrison, Valerie A. Paz-Soldan, Gonzalo M. Vazquez-Prokopec, William H. Elson, Patricia Barrera, Helvio Astete, Veronica Briesemeister, Mariana Leguia, Sarah A. Jenkins, Anna B. Kawiecki, T. Alex Perkins, Alun L. Lloyd, Robert D. Hontz, Christopher M. Barker, Alan L. Rothman, Thomas W. Scott.

**Methodology:** Amy C. Morrison, Valerie A. Paz-Soldan, Gonzalo M. Vazquez-Prokopec, Louis Lambrechts, Patricia Barrera, Helvio Astete, Veronica Briesemeister, Mariana Leguia, Sarah A. Jenkins, Kanya C. Long, Anna B. Kawiecki, Robert C. Reiner, Jr., Lance A. Waller, Steven T. Stoddard, Christopher M. Barker, Uriel Kitron, John P. Elder, Alan L. Rothman, Thomas W. Scott.

**Project administration:** Amy C. Morrison, Valerie A. Paz-Soldan, Gonzalo M. Vazquez-Prokopec, Veronica Briesemeister, Sarah A. Jenkins, Robert D. Hontz, Christopher M. Barker, Uriel Kitron, John P. Elder, Alan L. Rothman, Thomas W. Scott.

**Resources:** Amy C. Morrison, Valerie A. Paz-Soldan, Gonzalo M. Vazquez-Prokopec, Thomas W. Scott.

**Software:** Anna B. Kawiecki, Christopher M. Barker.

**Supervision:** Amy C. Morrison, Valerie A. Paz-Soldan, Gonzalo M. Vazquez-Prokopec, Patricia Barrera, Helvio Astete, Veronica Briesemeister, Sarah A. Jenkins, Robert D. Hontz, Christopher M. Barker, Alan L. Rothman, Thomas W. Scott.

**Validation:** Amy C. Morrison, William H. Elson, Helvio Astete, Mariana Leguia, Alan L. Rothman.

**Visualization:** Gonzalo M. Vazquez-Prokopec.

**Writing – original draft:** Amy C. Morrison, Thomas W. Scott.

**Writing – review & editing:** Amy C. Morrison, Valerie A. Paz-Soldan, Gonzalo M. Vazquez-Prokopec, Louis Lambrechts, William H. Elson, Patricia Barrera, Helvio Astete, Veronica Briesemeister, Mariana Leguia, Sarah A. Jenkins, Kanya C. Long, Anna B. Kawiecki, Robert C. Reiner, Jr., T. Alex Perkins, Alun L. Lloyd, Lance A. Waller, Robert D. Hontz, Steven T. Stoddard, Christopher M. Barker, Uriel Kitron, John P. Elder, Alan L. Rothman, Thomas W. Scott.

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
