## [Decision Letter · Decision Letter 0]

28 Jun 2022

PONE-D-22-01362

Quantifying Heterogeneities in Arbovirus Transmission: A Prospective Longitudinal Study of Dengue and Zika Viruses in Iquitos, Peru

PLOS ONE

Dear Dr. Morrison,

Thank you for submitting your manuscript to PLOS ONE. After careful consideration, we feel that it has merit but does not fully meet PLOS ONE’s publication criteria as it currently stands. Therefore, we invite you to submit a revised version of the manuscript that addresses the points raised during the review process.

Reviewers think that after minor modifications your manuscript has a chance to be accepted. Please try to respond to all queries raised by the reviewers.

We look forward to receiving your revised manuscript.

Kind regards,

Luciano Andrade Moreira, PhD

Academic Editor

PLOS ONE

“This study was funded by the U.S. National Institute of Allergy and Infectious Diseases (NIH/NIAID) award number P01 AI098670 (TWS). Further support was provided by Bill and Melinda Gates Foundation (BMGF) to the University of Notre Dame (Grant# OPP1081737) (NA), the Defense Threat Reduction Agency (DTRA) (RDH), Military Infectious Disease Research Program (MIDRP, S0520_15_LI and S0572_17_LI), and the Armed Forces Health Surveillance Branch Global Emerging Infections Systems research program (GEIS) ProMIS ID: 20160390169, P0090_17_N6_1.1.1, P0106_18_N6_01.01, and P0143_19_N6.”

4. Thank you for stating the following in the Funding Section of your manuscript:

“This study was funded by the U.S. National Institute of Allergy and Infectious Diseases (NIH/NIAID) award number P01 AI098670. Further support was provided by Bill and Melinda Gates Foundation (BMGF) to the University of Notre Dame (Grant# OPP1081737), the Defense Threat Reduction Agency (DTRA), Military Infectious Disease Research Program (MIDRP, S0520_15_LI and S0572_17_LI), and the Armed Forces Health Surveillance Branch Global Emerging Infections Systems research program (GEIS) ProMIS ID: 20160390169, P0090_17_N6_1.1.1, P0106_18_N6_01.01, and P0143_19_N6.”

“This study was funded by the U.S. National Institute of Allergy and Infectious Diseases (NIH/NIAID) award number P01 AI098670 (TWS). Further support was provided by Bill and Melinda Gates Foundation (BMGF) to the University of Notre Dame (Grant# OPP1081737) (NA), the Defense Threat Reduction Agency (DTRA) (RDH), Military Infectious Disease Research Program (MIDRP, S0520_15_LI and S0572_17_LI), and the Armed Forces Health Surveillance Branch Global Emerging Infections Systems research program (GEIS) ProMIS ID: 20160390169, P0090_17_N6_1.1.1, P0106_18_N6_01.01, and P0143_19_N6.”

6. We note that you have indicated that data from this study are available upon request. PLOS only allows data to be available upon request if there are legal or ethical restrictions on sharing data publicly. For more information on unacceptable data access restrictions, please see http://journals.plos.org/plosone/s/data-availability#loc-unacceptable-data-access-restrictions.

7. Please note that in order to use the direct billing option the corresponding author must be affiliated with the chosen institute. Please either amend your manuscript to change the affiliation or corresponding author, or email us at plosone@plos.org with a request to remove this option.

8. One of the noted authors is a group or consortium [Full list of names listed in the manuscript under Project Dengue Group]. In addition to naming the author group, please list the individual authors and affiliations within this group in the acknowledgments section of your manuscript. Please also indicate clearly a lead author for this group along with a contact email address.

9. Your ethics statement should only appear in the Methods section of your manuscript. If your ethics statement is written in any section besides the Methods, please move it to the Methods section and delete it from any other section. Please ensure that your ethics statement is included in your manuscript, as the ethics statement entered into the online submission form will not be published alongside your manuscript.

10. We note that [Figures 2 and 4] in your submission contain [map/satellite] images which may be copyrighted. All PLOS content is published under the Creative Commons Attribution License (CC BY 4.0), which means that the manuscript, images, and Supporting Information files will be freely available online, and any third party is permitted to access, download, copy, distribute, and use these materials in any way, even commercially, with proper attribution. For these reasons, we cannot publish previously copyrighted maps or satellite images created using proprietary data, such as Google software (Google Maps, Street View, and Earth). For more information, see our copyright guidelines: http://journals.plos.org/plosone/s/licenses-and-copyright.

a. You may seek permission from the original copyright holder of Figure 2 and 4 to publish the content specifically under the CC BY 4.0 license. 

Reviewers' comments:

Reviewer's Responses to Questions

**Comments to the Author**

1. Is the manuscript technically sound, and do the data support the conclusions?

Reviewer #1: Yes

Reviewer #2: Partly

2. Has the statistical analysis been performed appropriately and rigorously? 

Reviewer #1: Yes

Reviewer #2: N/A

3. Have the authors made all data underlying the findings in their manuscript fully available?

Reviewer #1: Yes

Reviewer #2: No

4. Is the manuscript presented in an intelligible fashion and written in standard English?

Reviewer #1: Yes

Reviewer #2: Yes

5. Review Comments to the Author

Reviewer #1: The project is very ambitious, well written and with a very experienced team. I believe it has a good chance of making great contributions to the knowledge of dengue. It must undergo a small English review before being published.

Reviewer #2: In this paper the authors describe an ambitious and comprehensive five-year program of work in Iquitos, Peru, that builds on two previous decades of dengue research in the city by the lead authors, and was designed to fill knowledge gaps in understanding the full spectrum of dengue virus infection outcomes and how these influence DENV transmission dynamics. The research program comprised integrated workstreams examining human-to-mosquito infectiousness, subjective and objective measurements of disease severity and quality of life measures, household cluster investigations to identify inapparent infections and understand spatial patterns of transmission, and analyses of the interaction of infectiousness and disease severity with human mobility, all unpinned by a combination of community-based active surveillance and clinic-based passive surveillance to detect and enrol viraemic dengue patients of all disease severities. The significance of this research program, and the quality and importance of the research findings it has produced, is not in question. What is not as clear as it could be, however, is the overall intention and the execution of this paper itself. Some relatively minor changes to the framing of the manuscript should help to resolve this, as suggested below.

Major comments

1. From the outset, starting with the title itself, it is not clear to the reader whether the focus of the paper is on i) reporting the structure, methodology and significance of the research program, ii) primary reporting of research findings, vs iii) a synthesis or review of research findings that have already been published. Throughout, the manuscript jumps between the first and third of these, but overall seems to be primarily focused on communicating the strengths of the research program's structure and methods, and the resulting synergies between the workstreams. This disconnect is most apparent in the closing sentence of the introduction, the closing paragraph of the 'Significance' section, and the closing paragraph of the manuscript, where the authors refer to expectations and hopes that the results "will have public health and basic science implications" and "will reveal new perspectives on processes in transmission of DENV...and provide key missing information for improving the design of dengue prevention strategies". This language is confusing as it implies that the research is yet to be conducted and the results are not yet known.

Given that a lot of the research findings have already been produced and published, it would be more meaningful to comment on what some of the public health and basic science implications of the key findings actually are.

Or alternatively (and more simply) the authors could make it explicit early on and throughout that results of this research have previously been published (with citations), and that the objective of this paper is to give a holistic view of the structure and methodology of the integrated research program, and how its design and implementation facilitated the achievement of the research objectives. In this case, the authors should reframe the introduction and conclusion of the paper to focus primarily on the methodological strengths of the integrated research program (as done in the penultimate paragraph of the paper) to address the knowledge gaps in dengue transmission and control, without focusing so heavily on the significance of the research findings themselves - since these findings aren't made known to the reader.

2. The title doesn't clearly communicate the scope of the paper, rather it presumably reflects the title of the research protocol. A more informative title would be something along the lines of 'An integrated program of entomological, epidemiological and clinical research to understand heterogeneities in arbovirus transmission in Iquitos, Peru'.

3. The abstract describes the objectives and activities of the Iquitos research program, and creates the impression that research findings will be presented herein. Instead the abstract should reflect the objective, scope and conclusions of this paper, i.e. including an overview of the research methodology but with a focus on the significance of the integrated projects and cores, and what has been achieved through this program.

Minor comments

- Page 5, line 164: The failure of dengue control programs has also been attributed to the lack of an evidence base for their effective implementation, generated through robust study designs with epidemiological endpoints (Wilson et al 2015; Bowman et al 2016)

- Page 15, line 420: the mosquito direct feeding experiments should be described in text here with reference to Figure 3, or a citation given for where these direct feeding experimental methods have been described previously.

- Page 16, Figure 3: the abbreviation FFA (and probably also RT-PCR) needs to be defined in the figure legend

- Page 20, line 544: there is a citation [3] here that is non-sequential with other citations.

- Page 20, line 548: this paragraph should have a new subheading reflecting its focus on highlighting synergies between the program workstreams (not 'Analysis plan and data integration', as it currently falls under).

- Page 20, line 549: P01 program is not familiar terminology for many readers. 'Research program' instead?

- Page 20, line 554: 'were also be used' - wording needs fixing

6. PLOS authors have the option to publish the peer review history of their article (what does this mean?). If published, this will include your full peer review and any attached files.

Reviewer #1: **Yes: **LUCIANO PAMPLONA DE GOES CAVALCANTI

Reviewer #2: No

---

## [Author Response · Author response to Decision Letter 0]

23 Jul 2022

Reviewer #1: The project is very ambitious, well written and with a very experienced team. I believe it has a good chance of making great contributions to the knowledge of dengue. It must undergo a small English review before being published.

RESPONSE: We reviewed and edited the manuscript for grammar and spelling.

Reviewer #2: In this paper the authors describe an ambitious and comprehensive five-year program of work in Iquitos, Peru, that builds on two previous decades of dengue research in the city by the lead authors, and was designed to fill knowledge gaps in understanding the full spectrum of dengue virus infection outcomes and how these influence DENV transmission dynamics. The research program comprised integrated workstreams examining human-to-mosquito infectiousness, subjective and objective measurements of disease severity and quality of life measures, household cluster investigations to identify inapparent infections and understand spatial patterns of transmission, and analyses of the interaction of infectiousness and disease severity with human mobility, all unpinned by a combination of community-based active surveillance and clinic-based passive surveillance to detect and enrol viraemic dengue patients of all disease severities. The significance of this research program, and the quality and importance of the research findings it has produced, is not in question. What is not as clear as it could be, however, is the overall intention and the execution of this paper itself. Some relatively minor changes to the framing of the manuscript should help to resolve this, as suggested below.

Major comments

1. From the outset, starting with the title itself, it is not clear to the reader whether the focus of the paper is on i) reporting the structure, methodology and significance of the research program, ii) primary reporting of research findings, vs iii) a synthesis or review of research findings that have already been published. Throughout, the manuscript jumps between the first and third of these, but overall seems to be primarily focused on communicating the strengths of the research program's structure and methods, and the resulting synergies between the workstreams. This disconnect is most apparent in the closing sentence of the introduction, the closing paragraph of the 'Significance' section, and the closing paragraph of the manuscript, where the authors refer to expectations and hopes that the results "will have public health and basic science implications" and "will reveal new perspectives on processes in transmission of DENV...and provide key missing information for improving the design of dengue prevention strategies". This language is confusing as it implies that the research is yet to be conducted and the results are not yet known.

RESPONSE. We made the following changes to clarify that the objective of the manuscript is option #1 stated above. It is not intended to include primary research findings. As noted in option #3, our manuscript provides some synthesis of previous research findings from our group, but not from this project. This project represented what our research group concluded was our next logical research priority based on our most recent research findings. In the revised manuscript we made modifications to explicitly state and clarify this point. Please see our responses to more specific comments below.

Given that a lot of the research findings have already been produced and published, it would be more meaningful to comment on what some of the public health and basic science implications of the key findings actually are.

Or alternatively (and more simply) the authors could make it explicit early on and throughout that results of this research have previously been published (with citations), and that the objective of this paper is to give a holistic view of the structure and methodology of the integrated research program, and how its design and implementation facilitated the achievement of the research objectives. In this case, the authors should reframe the introduction and conclusion of the paper to focus primarily on the methodological strengths of the integrated research program (as done in the penultimate paragraph of the paper) to address the knowledge gaps in dengue transmission and control, without focusing so heavily on the significance of the research findings themselves - since these findings aren't made known to the reader.

RESPONSE. We appreciate the reviewer’s comments and borrowing some of his/her suggested language. In the revised manuscript we now include a separate section called rational and end the introduction section by clearly stating the objectives of the manuscript. Please see the modified text below.

SEE LINES 120-132, WITH THE KEY ADDED TEXT BEING.

This publication describes our large and complex research program that followed our human movement studies. It is intended to be a single unified source that can be referenced in subsequent publications. Our aim is to provide a holistic overview of the integrated structure and methodology that we used to achieve our research objectives. Data collection for the project ended in 2019. Due to delays associated with the COVID-19 pandemic some data analysis and laboratory testing are ongoing. Although we previously published project associated reports (i.e., spectrum of dengue illness experienced in Iquitos [10] and impact of dengue on human mobility [11,12] and quality of life measures[13]) in this publication we do not focus on project related data. Instead, we limit our presentation to the overall rational and methodology of the study and how the project was designed to directly address key knowledge gaps in the understanding of DENV transmission.

WE ELIMINATED THE OUTLOOK SECTION AS SUGGESTED ABOVE, BUT ADDED SOME NEW TEXT EMPHASIZING PROJECT SYNERGIES IN A NEW SECTION CALLED “PROGRAMMATIC SYNERGIES” (Lines 540-570). Much of this text is new.

2. The title doesn't clearly communicate the scope of the paper, rather it presumably reflects the title of the research protocol. A more informative title would be something along the lines of 'An integrated program of entomological, epidemiological and clinical research to understand heterogeneities in arbovirus transmission in Iquitos, Peru'.

Response. The title has been modified to the following: Quantifying heterogeneities in arbovirus transmission: Description of the rational and methodology for a prospective longitudinal study of dengue and Zika virus transmission in Iquitos, Peru (2014-2019). 

3. The abstract describes the objectives and activities of the Iquitos research program, and creates the impression that research findings will be presented herein. Instead the abstract should reflect the objective, scope and conclusions of this paper, i.e. including an overview of the research methodology but with a focus on the significance of the integrated projects and cores, and what has been achieved through this program.

RESPONSE. Abstract (Lines 63-96), new text is shown in blue: Only in the last 20 years have there been significant efforts to carry out comprehensive longitudinal dengue studies. This manuscript provides the rational and comprehensive, integrated description of the methodology for a five-year longitudinal cohort study based in the tropical city of Iquitos, in the heart of the Peruvian Amazon. Primary data collection for this study was completed in 2019. Although some manuscripts have been published to date, our principal objective here is to support subsequent publications by describing in detail the structure, methodology, and significance of a specific research program. Our project was designed to study people across the entire continuum of disease, with the ultimate goal of quantifying heterogeneities in human variables that affect DENV transmission dynamics and prevention. Because our study design is applicable to other Aedes transmitted viruses, we used it to gain insights into Zika virus (ZIKV) transmission when during the project period ZIKV was introduced and circulated in Iquitos. Our prospective contact cluster investigation design was initiated by detection a person with a symptomatic DENV infection and then followed that person’s immediate contacts. This allowed us to monitor individuals at high risk of DENV infection, including people with clinically inapparent and mild disease that are otherwise difficult to detect. We aimed

Minor comments

- Page 5, line 164: The failure of dengue control programs has also been attributed to the lack of an evidence base for their effective implementation, generated through robust study designs with epidemiological endpoints (Wilson et al 2015; Bowman et al 2016)

Response: We agree. Lines 175-178 now read as follows.

Unsuccessful programs are often attributed to a lack of resources, lack of political will, ineffective implementation [1,25], and the absence of a rigorous evidence base for available methods that was generated through robust study designs with epidemiological endpoints[26,27]. Please note that we included the suggested references.

- Page 15, line 420: the mosquito direct feeding experiments should be described in text here with reference to Figure 3, or a citation given for where these direct feeding experimental methods have been described previously.

Response (see lines 434-444). We now reference Figure 3 on line 435 and the appropriate supplementary information sections (Lines 438-439) and provide an appropriate reference for the direct feeding procedures (reference 50 on line 443).

- Page 16, Figure 3: the abbreviation FFA (and probably also RT-PCR) needs to be defined in the figure legend

Response: Done.

- Page 20, line 544: there is a citation [3] here that is non-sequential with other citations.

Response: Corrected.

- Page 20, line 548: this paragraph should have a new subheading reflecting its focus on highlighting synergies between the program workstreams (not 'Analysis plan and data integration', as it currently falls under).

Response: We added the subtitle: Programmatic Synergies. This section includes some new text.

- Page 20, line 549: P01 program is not familiar terminology for many readers. 'Research program' instead?

Response: We removed the reference to P01 program (Lines 538-539). 

- Page 20, line 554: 'were also be used' - wording needs fixing

Response: Done.

---

## [Editor Report · Decision Letter 1]

16 Aug 2022

Quantifying heterogeneities in arbovirus transmission: Description of the rational and methodology for a prospective longitudinal study of dengue and Zika virus transmission in Iquitos, Peru (2014-2019)

PONE-D-22-01362R1

Dear Dr. Morrison,

We’re pleased to inform you that your manuscript has been judged scientifically suitable for publication and will be formally accepted for publication once it meets all outstanding technical requirements.

Kind regards,

Luciano Andrade Moreira, PhD

Academic Editor

PLOS ONE
---

## [Editor Report · Acceptance letter]

23 Aug 2022

PONE-D-22-01362R1 

Quantifying heterogeneities in arbovirus transmission: Description of the rational and methodology for a prospective longitudinal study of dengue and Zika virus transmission in Iquitos, Peru (2014-2019) 

Dear Dr. Morrison:

I'm pleased to inform you that your manuscript has been deemed suitable for publication in PLOS ONE. Congratulations! Your manuscript is now with our production department. 

Kind regards, 

on behalf of

Dr. Luciano Andrade Moreira 

Academic Editor

PLOS ONE